# Recursive Aggregation and Its Fusion Process for Intuitionistic Fuzzy Numbers Based on Non-Additive Measure

**Yongfu Shi** [1,*] **and Zengtai Gong** [2,*]

1   State-Owned Assets Management Office, Northwest Normal University, Lanzhou 730070, China
2   College of Mathematics and Statistics, Northwest Normal University, Lanzhou 730070, China;
*   Correspondence: yfshinwnu@163.com or shiyongfu@nwnu.edu.cn (Y.S.); zt-gong@163.com (G.Z.); Tel.: +86-13109371658 (Y.S.)

**Abstract:** In this paper, the recursive aggregation of OWA operators for intuitionistic fuzzy numbers (IFN) based on a non-additive measure (NAM) with $\sigma - \lambda$ rules is constructed and investigated in light of the $\sigma - \lambda$ rules of a non-additive measure (NAM). Additionally, an integrator is designed by drawing on the genetic algorithm and the process of calculation is elaborated by an example.

**Keywords:** OWA operators; intuitionistic fuzzy numbers (IFN); non-additive measure (NAM); recursive aggregation

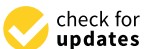

## 1. Introduction

Since the concept of ordered weighted averaging (OWA) operators was initiated by Yager in 1988 [1], it has received wide application in various domains, including decision expert systems, market surveys, neural networks, analysis, fuzzy logic control, etc. [2–4]. In 2005, Yager further put forward the recursive forms of OWA operators [5]. Its fundamental basic idea is to directly derive the aggregated result of $n$ data by using the aggregated result of $n - 1$ data while keeping the orness level unchanged. However, the attribute indexes are mostly related with each other in the process of deriving aggregation results of $n$ data using the existing aggregation results of $n - 1$ data. The main reason lies in the fact that the existing recursive forms of OWA operators are established on the base of the classical probability measure and Lebesgue integral integration operator [6]. That is to say, the weight vector is not able to be measured independently, and it may also not meet the countable additivity of the classical probability measure. Fortunately, fuzzy measure, defined by Sugeno in 1974, can be utilized to depict the correlations of attribute indexes [7–10]. Meanwhile, aggregation functions and aggregation operators are investigated by many researchers recently [11–14]. Because of the fuzziness and uncertainty of actual decision issues, the evaluation values involved in the decision process are not always expressed as crisp numbers. However, Intuitionistic fuzzy number (IFN) is a critical tool for settling imprecise information [15,16] and could offer the membership degree and the nonmembership degree simultaneously. Thus, IFN performs more flexibly and efficiently than a traditional fuzzy set in addressing uncertainty. In this work, recursive aggregation of OWA operators for IFN based on a non-additive measure (NAM) with $\sigma - \lambda$ rules is put forward and researched.

The rest of this work is organized as follows. In Section 2, we review the non-additive measure (NAM) with $\sigma - \lambda$ rules and IFN and propose an OWA operator for IFN based on a NAM with $\sigma - \lambda$ rules. In Section 3, we derive the recursive forms of OWA operators for IFN according to NAM with $\sigma - \lambda$ rules while keeping the orness grade unchanged. In Section 4, the procedure for integrator design in recursive aggregation is designed, and the process of calculation is demonstrated by an example.

## 2. Preliminaries

**Definition 1** ([7–10]). *Let $X$ be a nonempty set and $\mathscr{A}$ a $\sigma-$ algebra on the $X$. A set function $\mu : A \rightarrow [0, +\infty]$ is called a fuzzy measure if*
*(1) $\mu(\varnothing) = 0$;*
*(2) $\mu(X) = 1$;*
*(3) For each $A$ and $B \in \mathscr{A}$ such that $A \subseteq B$, $\mu(A) \leq \mu(B)$.*
*A fuzzy measure $\mu$ is called a Sugeno measure if $\mu$ satisfies $\sigma - \lambda$ rules, briefly denoted as $g_\lambda$. The fuzzy measure shown in this paper is a Sugeno measure.*

**Definition 2** ([7–10]). *$g_\lambda$ is called a fuzzy measure based on $\sigma - \lambda$ rules if*

$$g_\lambda \left( \bigcup_{i=1}^{\infty} A_i \right) = \begin{cases} \dfrac{1}{\lambda} \left\{ \prod_{i=1}^{\infty} [1 + \lambda g_\lambda(A_i)] - 1 \right\}, & \lambda \neq 0, \\ \displaystyle\sum_{i=1}^{\infty} g_\lambda(A_i), & \lambda = 0, \end{cases}$$

*where $\lambda \in (-\frac{1}{\sup\mu}, \infty) \bigcup \{0\}$, $\{A_i\} \subset \mathscr{A}$, $A_i \cap A_j = \varnothing$ for all $i, j = 1, 2, \cdots$ and $i \neq j$.*

Particularly, if $\lambda = 0$, then $g_\lambda$ is a classic probability measure.
In Definition 2, if $n = 2$, then

$$\mu(A \cup B) = \begin{cases} \mu(A) + \mu(B) + \lambda\mu(A)\mu(B), & \lambda \neq 0, \\ \mu(A) + \mu(B), & \lambda = 0. \end{cases}$$

If $X$ is a finite set, for any subset $A$ of $X$, then

$$g_\lambda(A) = \begin{cases} \dfrac{1}{\lambda} \left\{ \prod_{x \in A} [1 + \lambda g_\lambda(\{x\})] - 1 \right\}, & \lambda \neq 0, \\ \displaystyle\sum_{x \in A} g_\lambda(\{x\}), & \lambda = 0. \end{cases}$$

If $X$ is a finite set, then the parameter $\lambda$ of a regular Sugeno measure based on $\sigma - \lambda$ rules is determined by the equation

$$1 + \lambda = \prod_{i=1}^{n} (1 + \lambda g_\lambda(x_i)).$$

**Definition 3** ([15,16]). *(IFN$_s$) Suppose $X$ be a universe of discourse. An INF in $A$ over $X$ is expressed by:*
$$A = \{ \langle x, \mu_A(x), \nu_A(x) \rangle | x \in X \},$$
*where*
$$\mu_A \rightarrow [0, 1], \nu_A \rightarrow [0, 1],$$
*with the condition $0 \leq \mu(x) + \nu(x) \leq 1, \forall x \in X$. $\mu(x)$, $\nu(x)$ and $\pi(x) = 1 - \mu(x) - \nu(x)$ denote the membership function, the non-membership function, and hesitation functions of the element $x$ to the set $A$, respectively. The class $\tilde{\mathfrak{E}}$ represents the set of all INFs in $A$.*

**Definition 4** ([16]). *(IFN$_s$ Score Function) Suppose $\tilde{\alpha} = (\mu, \nu)$ be an IFN, a score function $S(\tilde{\alpha})$ of an INF can be represented below*

$$S(\tilde{\alpha}) = \frac{\mu_{\tilde{\alpha}} - \nu_{\tilde{\alpha}}}{2}, \ S(\tilde{\alpha}) \in [-1, 1],$$

*to evaluate the degree of the score of the intuitionistic fuzzy number $\tilde{\alpha}$, where $S(\tilde{\alpha}) \in [-1, 1]$.*

*(IFN$_s$ Accuracy Function). Suppose $\tilde{\alpha} = (\mu, \nu)$ be an intuitionistic fuzzy number, an accuracy function $H(\tilde{\alpha})$ of an IFN can be defined below*

$$H(\tilde{\alpha}) = \frac{\mu_{\tilde{\alpha}} + \nu_{\tilde{\alpha}}}{2}, H(\tilde{\alpha}) \in [-1, 1],$$

*to evaluate the degree of accuracy of the IFN $\tilde{\alpha}$, where $H(\tilde{\alpha}) \in [-1, 1]$.*

In the light of the two mentioned functions in Definition 4, Xu and Yager proposed the order relation below [16]:

(1) If $S(\tilde{\alpha_1}) < S(\tilde{\alpha_2})$, then $\tilde{\alpha_1} < \tilde{\alpha_2}$;
(2) If $S(\tilde{\alpha_1}) = S(\tilde{\alpha_2})$, then
   (i) If $H(\tilde{\alpha}_1) = H(\tilde{\alpha}_2)$, then $\tilde{\alpha}_1 = \tilde{\alpha}_2$;
   (ii) If $H(\tilde{\alpha}_1) < H(\tilde{\alpha}_2)$, then $\tilde{\alpha}_1 < \tilde{\alpha}_2$;
   (iii) If $H(\tilde{\alpha}_1) > H(\tilde{\alpha}_2)$, then $\tilde{\alpha}_1 > \tilde{\alpha}_2$.

**Definition 5** ([16])**.** *Let $\tilde{\alpha}_1 = (\mu_{\tilde{\alpha}_1}, \nu_{\tilde{\alpha}_1})$ and $\tilde{\alpha}_2 = (\mu_{\tilde{\alpha}_2}, \nu_{\tilde{\alpha}_2})$ be two IFNs, then*
   *(1) $\tilde{\alpha}_1 \oplus \tilde{\alpha}_2 = (\mu_{\tilde{\alpha}_1} + \mu_{\tilde{\alpha}_2} - \mu_{\tilde{\alpha}_1}\mu_{\tilde{\alpha}_2}, \nu_{\tilde{\alpha}_1}\nu_{\tilde{\alpha}_2})$;*
   *(2) $\tilde{\alpha}_1 \otimes \tilde{\alpha}_2 = (\mu_{\tilde{\alpha}_1}\mu_{\tilde{\alpha}_2}, \nu_{\tilde{\alpha}_1} + \nu_{\tilde{\alpha}_2} - \nu_{\tilde{\alpha}_1}\nu_{\tilde{\alpha}_2})$;*
   *(3) $\lambda\tilde{\alpha} = (1 - (1 - \mu_{\tilde{\alpha}})^{\lambda}, \nu_{\tilde{\alpha}}^{\lambda}), \lambda > 0$;*
   *(4) $\tilde{\alpha}^{\lambda} = (\mu_{\tilde{\alpha}}^{\lambda}, 1 - (1 - \nu_{\tilde{\alpha}})^{\lambda}), \lambda > 0$.*

## 3. The OWA Operator for Intuitionistic Fuzzy Numbers (IFNs) on a Non-Additive Measure (NAM) with $\sigma - \lambda$ Rules

In the following discussion, we will always default to $g_{\lambda}(A_0) = 0$ unless otherwise specified.

**Definition 6.** *Let $g_{\lambda}$ be a fuzzy measure satisfying $\sigma - \lambda$ rules, $A_i = \{x_1, x_2, \cdots, x_i\}$, $i = 1, 2, \cdots, n$. An OWA operator of dimension $n$ for INFs based on a NAM with $\sigma - \lambda$ rules is a mapping $\tilde{F}_n : \tilde{\mathfrak{E}}_1 \times \tilde{\mathfrak{E}}_2 \times \cdots \times \tilde{\mathfrak{E}}_n \to \tilde{\mathfrak{E}}$ defined as*

$$\tilde{F}_n(\tilde{\alpha}_1, \tilde{\alpha}_2, \cdots, \tilde{\alpha}_n) = \bigoplus_{i=1}^{n}(g_{\lambda}(A_i) - g_{\lambda}(A_{i-1})) \cdot \tilde{\beta}_i, \tag{1}$$

*where $\tilde{\beta}_i$ is the i-th largest value out of $\tilde{\alpha} = (\tilde{\alpha}_1, \tilde{\alpha}_2, \cdots, \tilde{\alpha}_n)$ (i.e., $\tilde{\beta}_1 \geq \tilde{\beta}_2 \geq \cdots \geq \tilde{\beta}_n$).*

By Definitions 5 and 6, we can easily obtain the result below.

**Theorem 1.** *Let $g_{\lambda}$ be fuzzy measure satisfying $\sigma - \lambda$ rules, denote $A_i = \{x_1, x_2, \cdots, x_i\}$, $i = 1, 2, \cdots, n$, $\tilde{F}_n(\tilde{\alpha}_1, \tilde{\alpha}_2, \cdots, \tilde{\alpha}_n)$ be an OWA operator of dimension $n$ for IFNs based on a NAM with $\sigma - \lambda$ rules. Then,*

$$\tilde{F}_n(\tilde{\alpha}_1, \tilde{\alpha}_2, \cdots, \tilde{\alpha}_n) \tag{2}$$
$$= \bigoplus_{i=1}^{n}(1 - (1 - \mu_{\tilde{\beta}_i})^{g_{\lambda}(A_i)-g_{\lambda}(A_{i-1})}, \nu_{\tilde{\beta}_i}^{g_{\lambda}(A_i)-g_{\lambda}(A_{i-1})})$$
$$= (1 - \prod_{i=1}^{n}(1 - \mu_{\tilde{\beta}_i})^{g_{\lambda}(A_i)-g_{\lambda}(A_{i-1})}, \prod_{i=1}^{n}\nu_{\tilde{\beta}_i}^{g_{\lambda}(A_i)-g_{\lambda}(A_{i-1})}).$$

*That is to say,*

$$\mu_{\tilde{F}_n} = 1 - \prod_{i=1}^{n} (1 - \mu_{\tilde{\beta}_i})^{g_\lambda(A_i) - g_\lambda(A_{i-1})}, \tag{3}$$

$$\nu_{\tilde{F}_n} = \prod_{i=1}^{n} \nu_{\tilde{\beta}_i}^{g_\lambda(A_i) - g_\lambda(A_{i-1})},$$

*where $\tilde{\beta}_i = (\mu_{\tilde{\beta}_i}, \nu_{\tilde{\beta}_i})$ is the i-th largest value out of $\tilde{\alpha} = (\tilde{\alpha}_1, \tilde{\alpha}_2, \cdots, \tilde{\alpha}_n)$ (i.e., $\tilde{\beta}_1 \geq \tilde{\beta}_2 \geq \cdots \geq \tilde{\beta}_n$).*

**Proof.** By Definitions 5 and 6, we easily obtain the result below. As

$$\tilde{\beta}_1 \oplus \tilde{\beta}_2 = (\mu_{\tilde{\beta}_1} + \mu_{\tilde{\beta}_2} - \mu_{\tilde{\beta}_1}\mu_{\tilde{\beta}_2}, \nu_{\tilde{\beta}_1}\nu_{\tilde{\beta}_2}) = (1 - (1 - \mu_{\tilde{\beta}_1})(1 - \mu_{\tilde{\beta}_2}), \nu_{\tilde{\beta}_1}\nu_{\tilde{\beta}_2})$$

and

$$\tilde{\beta}_1 \oplus \tilde{\beta}_2 \oplus \tilde{\beta}_3 = (1 - (1 - \mu_{\tilde{\beta}_1})(1 - \mu_{\tilde{\beta}_2})(1 - \mu_{\tilde{\beta}_3}), \nu_{\tilde{\beta}_1}\nu_{\tilde{\beta}_2}\nu_{\tilde{\beta}_3}),$$

we have

$$\tilde{F}_n(\tilde{\alpha}_1, \tilde{\alpha}_2, \cdots, \tilde{\alpha}_n)$$
$$= \bigoplus_{i=1}^{n} (1 - (1 - \mu_{\tilde{\beta}_i})^{g_\lambda(A_i) - g_\lambda(A_{i-1})}, \nu_{\tilde{\beta}_i}^{g_\lambda(A_i) - g_\lambda(A_{i-1})})$$
$$= (1 - \prod_{i=1}^{n} (1 - \mu_{\tilde{\beta}_i})^{g_\lambda(A_i) - g_\lambda(A_{i-1})}, \prod_{i=1}^{n} \nu_{\tilde{\beta}_i}^{g_\lambda(A_i) - g_\lambda(A_{i-1})}).$$

It follows that

$$\mu_{\tilde{F}_n} = 1 - \prod_{i=1}^{n} (1 - \mu_{\tilde{\beta}_i})^{g_\lambda(A_i) - g_\lambda(A_{i-1})},$$

$$\nu_{\tilde{F}_n} = \prod_{i=1}^{n} \nu_{\tilde{\beta}_i}^{g_\lambda(A_i) - g_\lambda(A_{i-1})},$$

□

**Theorem 2.** *When $\lambda = 0$, and $\tilde{\beta}_i$ is an IFN, then the OWA operator for a series of INFs based on a NAM with $\sigma - \lambda$ rules would degenerate to the classic OWA operator form for a series of IFNs. In fact, based on countable additivity, we have*

$$\tilde{F}_n(\tilde{\alpha}_1, \tilde{\alpha}_2, \cdots, \tilde{\alpha}_n) = \bigoplus_{i=1}^{n} (g_\lambda(A_i) - g_\lambda(A_{i-1})) \cdot \tilde{\beta}_i \tag{4}$$
$$= g_1\tilde{\beta}_1 \bigoplus g_2\tilde{\beta}_2 \bigoplus \cdots \bigoplus g_n\tilde{\beta}_n$$
$$= \omega_1\tilde{\beta}_1 \bigoplus \omega_2\tilde{\beta}_2 \bigoplus \cdots \bigoplus \omega_n\tilde{\beta}_n$$
$$= \bigoplus_{i=1}^{n} (1 - (1 - \mu_{\tilde{\beta}_i})^{\omega_i}, \nu_{\tilde{\beta}_i}^{\omega_i})$$
$$= (1 - \prod_{i=1}^{n} (1 - \mu_{\tilde{\beta}_i})^{\omega_i}, \prod_{i=1}^{n} \nu_{\tilde{\beta}_i}^{\omega_i}).$$

*and*

$$\sum_{i=1}^{n} \omega_i = \sum_{i=1}^{n} (g_\lambda(A_i) - g_\lambda(A_{i-1})) = g_\lambda(A_n) = 1.$$

**Corollary 1.** *When $\lambda = 0$, and $\tilde{\beta}_i$ is a special IFN, namely, real number, $i = 1, 2, 3, \cdots$ then the OWA operator for a series of IFN based on a NAM with $\sigma - \lambda$ rules degenerates to the classic OWA operator in Reference [2].*

**Definition 7.** *Let $g_\lambda$ be fuzzy measure meeting $\delta - \lambda$ rules. Denote $A_i = \{x_1, x_2, \cdots, x_i\}$, $i = 1, 2, \cdots, n$, $A_0 = \emptyset$. The measure of orness involved with an OWA operator $\tilde{F}_n$ of dimension n for IFNs based on a NAM with $\sigma - \lambda$ rules can be defined by*

$$orness(g_\lambda^{(n)}) = \sum_{i=1}^{n} \frac{n-i}{n-1}(g_\lambda^{(n)}(A_i) - g_\lambda^{(n)}(A_{i-1})),$$

*The measure of andness associated with the OWA operator $\tilde{F}_n$ of dimension n for IFNs based on a NAM with $\sigma - \lambda$ rules can be further defined by*

$$andness(g_\lambda^{(n)}) = 1 - orness(g_\lambda^{(n)}) = \sum_{i=1}^{n} \frac{i-1}{n-1}(g_\lambda^{(n)}(A_i) - g_\lambda^{(n)}(A_{i-1})).$$

**Theorem 3.** *Let $g_\lambda$ be the fuzzy measure meeting $\delta - \lambda$ rules. Denote $A_i = \{x_1, x_2, \cdots, x_i\}$, $i = 1, 2, \cdots, n$, $A_0 = \emptyset$. $\tilde{F}_n$ is an OWA operator for IFNs based on a NAM with $\sigma - \lambda$ rules, then*
*(1) The measure of orness associated with an OWA operator $\tilde{F}_n$ of dimension n for IFNs based on a NAM with $\sigma - \lambda$ rules is defined as*

$$orness(g_\lambda^{(n)}) = \frac{1}{n-1} \sum_{i=1}^{n-1} g_\lambda^{(n)}(A_i), \tag{5}$$

*(2) The measure of andness associated with the OWA operator $\tilde{F}_n$ of dimension n for IFN based on a NAM with $\sigma - \lambda$ rules is defined as*

$$andness(g_\lambda^{(n)}) = 1 - orness(g_\lambda^{(n)}) = 1 - \frac{1}{n-1} \sum_{i=1}^{n-1} g_\lambda^{(n)}(A_i). \tag{6}$$

**Proof.**

$$\begin{aligned}
(1) \ orness(g_\lambda^{(n)}) &= \sum_{i=1}^{n} \frac{n-i}{n-1}(g_\lambda^{(n)}(A_i) - g_\lambda^{(n)}(A_{i-1})) \\
&= \frac{1}{n-1}[(n-1)(g_\lambda^{(n)}(A_1) - g_\lambda^{(n)}(A_0)) + (n-2)(g_\lambda^{(n)}(A_2) - g_\lambda^{(n)}(A_1)) \\
&+ \cdots + (g_\lambda^{(n)}(A_{n-1}) - g_\lambda^{(n)}(A_{n-2}))] \\
&= \frac{1}{n-1}(g_\lambda^{(n)}(A_1) + g_\lambda^{(n)}(A_2) + \cdots + g_\lambda^{(n)}(A_{n-1})) \\
&= \frac{1}{n-1} \sum_{i=1}^{n-1} g_\lambda^{(n)}(A_i),
\end{aligned}$$

$$(2)\ andness(g_\lambda^{(n)}) = \sum_{i=1}^{n} \frac{i-1}{n-1}(g_\lambda^{(n)}(A_i) - g_\lambda^{(n)}(A_{i-1}))$$

$$= \frac{1}{n-1}[(g_\lambda^{(n)}(A_2) - g_\lambda^{(n)}(A_1)) + 2(g_\lambda^{(n)}(A_3) - g_\lambda^{(n)}(A_2))$$

$$+ \cdots + (n-1)(g_\lambda^{(n)}(A_n) - g_\lambda^{(n)}(A_{n-1}))]$$

$$= \frac{1}{n-1}(-g_\lambda^{(n)}(A_1) - g_\lambda^{(n)}(A_2) - \cdots - g_\lambda^{(n)}(A_{n-1}) + (n-1))$$

$$= \frac{1}{n-1}(-\sum_{i=1}^{n-1} g_\lambda^{(n)}(A_i) + (n-1))$$

$$= 1 - \frac{1}{n-1}\sum_{i=1}^{n-1} g_\lambda^{(n)}(A_i).$$

The proof is complete. □

**Remark 1.** *When $\lambda = 0$, the measure of orness associated with an OWA operator $\tilde{F}_n$ of dimension $n$ for IFNs based on a NAM with $\sigma - \lambda$ rules degenerates to the classic case [5].*

**Remark 2.** *When $\lambda = 0$, and the weighting vector is $(1, 0, \cdots, 0)$, then $orness(g_\lambda^{(n)}) = 1$.*

**Remark 3.** *When $\lambda = 0$, and the weighting vector is $(0, 0, \cdots, 1)$, then $orness(g_\lambda^{(n)}) = 0$.*

**Remark 4.** *When $\lambda = 0$, and the weighting vector is $(\frac{1}{n}, \frac{1}{n}, \cdots, \frac{1}{n})$, then $orness(g_\lambda^{(n)}) = \frac{1}{2}$.*

## 4. Recursive Aggregation of the OWA Operator for Intuitionistic Fuzzy Numbers (INFs) Based on a Non-Additive Measure (NAM) with $\sigma - \lambda$ Rules

In this part, we will derive recursive aggregation of the OWA operator for INFs based on a NAM with $\sigma - \lambda$ rules under the condition that the orness grade remains unchanged.

**Lemma 1.** *Suppose $\tilde{\alpha}_1 = (\mu_{\tilde{\alpha}_1}, \nu_{\tilde{\alpha}_1})$ and $\tilde{\alpha}_2 = (\mu_{\tilde{\alpha}_2}, \nu_{\tilde{\alpha}_2})$ be two IFNs, and $\lambda_1, \lambda_2 \geq 0$, then*
*(1)$\lambda_1 \tilde{\alpha} \oplus \lambda_2 \tilde{\alpha} = (\lambda_1 + \lambda_2)\tilde{\alpha}$;*
*(2)$\lambda_1(\lambda_2 \tilde{\alpha}) = (\lambda_1 \lambda_2)\tilde{\alpha}$.*

**Theorem 4.** *Let $g_\lambda$ be the fuzzy measure meeting $\delta - \lambda$ rules. Denote $A_i = \{x_1, x_2, \cdots, x_i\}$, $i = 1, 2, \cdots, n$. $(g_\lambda^{(n)}(A_i) - g_\lambda^{(n)}(A_{i-1}))$ is the $i$-th element for the weighting vector of dimension $n$. $P_L^{(n)}$ denotes the correlation coefficient. The Left Recursive Form (LRF) of the OWA operator for IFNs based on a NAM with $\sigma - \lambda$ rules can be expressed as:*

$$\tilde{F}_n = P_L^{(n)} \cdot (\bigoplus_{i=1}^{n-1}(g_\lambda^{(n-1)}(A_i) - g_\lambda^{(n-1)}(A_{i-1})) \cdot \tilde{\beta}_i) \bigoplus (g_\lambda^{(n)}(A_n) - g_\lambda^{(n)}(A_{n-1})) \cdot \tilde{\beta}_n \quad (7)$$

$$= P_L^{(n)} \cdot \tilde{F}_{n-1} \bigoplus (1 - g_\lambda^{(n)}(A_{n-1})) \cdot \tilde{\beta}_n,$$

*where*

$$P_L^{(n)} = 1 - (1 - g_\lambda^{(n)}(A_{n-1})) = g_\lambda^{(n)}(A_{n-1}), \quad (8)$$
$$g_\lambda^{(n)}(A_i) - g_\lambda^{(n)}(A_{i-1}) = P_L^{(n)} \cdot (g_\lambda^{(n-1)}(A_i) - g_\lambda^{(n-1)}(A_{i-1})),$$
$$\sum_{i=1}^{n-1}(g_\lambda^{(n-1)}(A_i) - g_\lambda^{(n-1)}(A_{i-1})) = 1.$$

*For a fixed level of orness $\alpha$, we have*

$$P_L^{(n)} = \frac{(n-1)\alpha}{1 + \sum\limits_{i=1}^{n-2} g_\lambda^{(n-1)}(A_i)},$$

$$g_\lambda^{(n)}(A_1) = g_\lambda^{(n-1)}(A_1) \cdot P_L^{(n)},$$

$$\cdots$$

$$g_\lambda^{(n)}(A_{n-2}) = g_\lambda^{(n-1)}(A_{n-2}) \cdot P_L^{(n)},$$

$$g_\lambda^{(n)}(A_{n-1}) = P_L^{(n)}.$$

*Furthermore, $\tilde{\beta}_i$ is the i-th largest value out of $\tilde{\alpha} = (\tilde{\alpha}_1, \tilde{\alpha}_2, \cdots, \tilde{\alpha}_n)$ (i.e., $\tilde{\beta}_1 \geq \tilde{\beta}_2 \geq \cdots \geq \tilde{\beta}_n$).*

**Proof.** The simplest aggregation is for two elements, as

$$\tilde{F}_2 = (g_\lambda^{(2)}(A_1) - g_\lambda^{(2)}(A_0)) \cdot \tilde{\beta}_1 \bigoplus (g_\lambda^{(2)}(A_2) - g_\lambda^{(2)}(A_1)) \cdot \tilde{\beta}_2 = g_\lambda^{(2)}(A_1) \cdot \tilde{\beta}_1 \bigoplus (1 - g_\lambda^{(2)}(A_1)) \cdot \tilde{\beta}_2,$$

$$orness(g_\lambda^{(2)}) = g_\lambda^{(2)}(A_1) = \alpha, \; andness(g_\lambda^{(2)}) = 1 - g_\lambda^{(2)}(A_1) = 1 - \alpha.$$

Let us now consider the aggregation $\tilde{F}_3$. In this case,

$$orness(g_\lambda^{(3)}) = \frac{1}{2} \sum_{i=1}^{2} g_\lambda^{(3)}(A_i) = \alpha.$$

This leads to the system of independent equations

$$\begin{cases} g_\lambda^{(3)}(A_1) + g_\lambda^{(3)}(A_2) = 2\alpha, \\ (g_\lambda^{(3)}(A_1) - 0) + (g_\lambda^{(3)}(A_2) - g_\lambda^{(3)}(A_1)) + (g_\lambda^{(3)}(A_3) - g_\lambda^{(3)}(A_2)) = 1, \\ g_\lambda^{(3)}(A_1) = g_\lambda^{(2)}(A_1) \cdot P_L^{(3)}, \\ (g_\lambda^{(3)}(A_2) - g_\lambda^{(3)}(A_1)) = (1 - g_\lambda^{(2)}(A_1)) \cdot P_L^{(3)}. \end{cases}$$

The solution is

$$P_L^{(3)} = \frac{2\alpha}{1 + g_\lambda^{(2)}(A_1)},$$

$$g_\lambda^{(3)}(A_1) = g_\lambda^{(2)}(A_1) \cdot P_L^{(3)},$$

$$g_\lambda^{(3)}(A_2) = P_L^{(3)}.$$

More generally, in the case of $n$ arguments, we obtain the system of $n+1$ independent equations

$$\begin{cases} \sum\limits_{i=1}^{n-1} g_\lambda^{(n)}(A_i) = (n-1)\alpha, \\ \sum\limits_{i=1}^{n} (g_\lambda^{(n)}(A_i) - g_\lambda^{(n)}(A_{i-1})) = 1, \\ g_\lambda^{(n)}(A_1) = g_\lambda^{(n-1)}(A_1) \cdot P_L^{(n)}, \\ \cdots \\ g_\lambda^{(n)}(A_{n-1}) = g_\lambda^{(n-1)}(A_{n-1}) \cdot P_L^{(n)}, \end{cases}$$

whose solution is

$$P_L^{(n)} = \frac{(n-1)\alpha}{1 + \sum\limits_{i=1}^{n-2} g_\lambda^{(n-1)}(A_i)},$$

$$g_\lambda^{(n)}(A_1) = g_\lambda^{(n-1)}(A_1) \cdot P_L^{(n)},$$

$$\cdots$$

$$g_\lambda^{(n)}(A_{n-2}) = g_\lambda^{(n-1)}(A_{n-2}) \cdot P_L^{(n)},$$

$$g_\lambda^{(n)}(A_{n-1}) = P_L^{(n)}.$$

The proof is complete. □

**Theorem 5.** *Let $g_\lambda$ be the fuzzy measure meeting $\sigma - \lambda$ rules, and let $A_i = \{x_1, x_2, \cdots, x_i\}$, $i = 1, 2, \cdots, n$, $\tilde{F}_n(\tilde{\alpha}_1, \tilde{\alpha}_2, \cdots, \tilde{\alpha}_n)$ be an OWA operator of dimension n for IFNs based on a NAM with $\sigma - \lambda$ rules. $P_L^{(n)}$ denotes the correlation coefficient. $\tilde{F}_n = (\mu_{\tilde{F}_n}, \nu_{\tilde{F}_n})$, $\tilde{\beta}_n = (\mu_{\tilde{\beta}_n}, \nu_{\tilde{\beta}_n})$. Then,*

$$\tilde{F}_n = (1 - (1 - \mu_{\tilde{F}_{n-1}})^{P_L^{(n)}} (1 - \mu_{\tilde{\beta}_n})^{1 - g_\lambda^{(n)}(A_{n-1})}, \nu_{\tilde{F}_{n-1}}^{P_L^{(n)}} \nu_{\tilde{\beta}_n}^{1 - g_\lambda^{(n)}(A_{n-1})}). \tag{9}$$

*That is to say,*

$$\mu_{\tilde{F}_n} = 1 - (1 - \mu_{\tilde{F}_{n-1}})^{P_L^{(n)}} (1 - \mu_{\tilde{\beta}_n})^{1 - g_\lambda^{(n)}(A_{n-1})}, \tag{10}$$

$$\nu_{\tilde{F}_n} = \nu_{\tilde{F}_{n-1}}^{P_L^{(n)}} \nu_{\tilde{\beta}_n}^{1 - g_\lambda^{(n)}(A_{n-1})},$$

*where*

$$P_L^{(n)} = 1 - (1 - g_\lambda^{(n)}(A_{n-1})) = g_\lambda^{(n)}(A_{n-1}),$$

$$g_\lambda^{(n)}(A_i) - g_\lambda^{(n)}(A_{i-1}) = P_L^{(n)} \cdot (g_\lambda^{(n-1)}(A_i) - g_\lambda^{(n-1)}(A_{i-1})),$$

$$\sum_{i=1}^{n-1} (g_\lambda^{(n-1)}(A_i) - g_\lambda^{(n-1)}(A_{i-1})) = 1.$$

*For a fixed level of orness $\alpha$, we have*

$$P_L^{(n)} = \frac{(n-1)\alpha}{1 + \sum\limits_{i=1}^{n-2} g_\lambda^{(n-1)}(A_i)},$$

$$g_\lambda^{(n)}(A_1) = g_\lambda^{(n-1)}(A_1) \cdot P_L^{(n)},$$

$$\cdots$$

$$g_\lambda^{(n)}(A_{n-2}) = g_\lambda^{(n-1)}(A_{n-2}) \cdot P_L^{(n)},$$

$$g_\lambda^{(n)}(A_{n-1}) = P_L^{(n)}.$$

*Furthermore, $\tilde{\beta}_i = (\mu_{\tilde{\beta}_i}, \nu_{\tilde{\beta}_i})$ is the i-th largest value out of $\tilde{\alpha} = (\tilde{\alpha}_1, \tilde{\alpha}_2, \cdots, \tilde{\alpha}_n)$ (i.e., $\tilde{\beta}_1 \geq \tilde{\beta}_2 \geq \cdots \geq \tilde{\beta}_n$).*

**Remark 5.** *Similar to Theorems 4 and 5, the Reft Recursive Form (RRF) and the General Recursive Form (GRF) of the OWA operator for IFNs based on a NAM with $\sigma - \lambda$ rules can be discussed easily. However, the Recursive Aggregation (RA) refers to Reft Recursive Form (RRF) if it is not emphasized.*

**Remark 6.** *However, in practice, it is more suitable to settle some problems with a fixed level of orness $\alpha$. Thus, it is interesting to notice that $P_L^{(n)}$ depends on $n$ and $\alpha$, as*

$$P_L^{(n)} = P_L(n, \alpha) = \frac{(n-1)\alpha}{(n-2)\alpha + 1}. \tag{11}$$

## 5. Calculation of NAM and Fusion Process Design Based on Recursive Aggregation

**Theorem 6.** *Let $\tilde{\alpha}_i = (\mu_{\tilde{\alpha}_i}, v_{\tilde{\alpha}_i})(i = 1, 2)$ be two intuitionistic fuzzy numbers. The distance measure of IFNs $\tilde{\alpha}_1$ and $\tilde{\alpha}_2$, referring to [17], is defined by*

$$D^2(\tilde{\alpha}_1, \tilde{\alpha}_2) = \frac{1}{2}((\mu_{\tilde{\alpha}_1} - \mu_{\tilde{\alpha}_2})^2 + (v_{\tilde{\alpha}_1} - v_{\tilde{\alpha}_2})^2 + (\pi_{\tilde{\alpha}_1} - \pi_{\tilde{\alpha}_2})^2).$$

An OWA operator for IFNs based on a NAM with $\sigma - \lambda$ rules is a multiple input and single output model. By solving the model, we can obtain necessary data.

Let $X = \{x_1, x_2, \cdots, x_k, x_{k+1}, \cdots, x_n\}, n \geq 2$ be a set of attributes, and $A = \{\alpha_1, \alpha_2, \cdots, \alpha_m\}$ be a set of objects. Y is the given target. Table 1 shows the evaluation information, where $\tilde{\alpha}_{ij} = (\mu_{\tilde{\alpha}_{ij}}, v_{\tilde{\alpha}_{ij}})$ is a value for *i*-th object with respect to attribute $x_j$, and $\tilde{E}_i = (\mu_{\tilde{E}_i}, \mu_{\tilde{E}_i})$ is an evaluation value.

**Table 1.** Information evaluation table.

|            | $x_1$              | $x_2$              | $\cdots$ | $x_k$              | $y$         |
| ---------- | ------------------ | ------------------ | -------- | ------------------ | ----------- |
| $\alpha_1$   | $\tilde{\alpha}_{11}$ | $\tilde{\alpha}_{12}$ |          | $\tilde{\alpha}_{1k}$ | $\tilde{E}_1$ |
| $\alpha_2$   | $\tilde{\alpha}_{21}$ | $\tilde{\alpha}_{22}$ | $\cdots$ | $\tilde{\alpha}_{2k}$ | $\tilde{E}_2$ |
| $\cdots$     | $\cdots$           | $\cdots$           | $\cdots$ | $\cdots$           | $\cdots$    |
| $\alpha_m$   | $\tilde{\alpha}_{m1}$ | $\tilde{\alpha}_{m2}$ | $\cdots$ | $\tilde{\alpha}_{mk}$ | $\tilde{E}_m$ |

Below, we present a procedure for the calculation of the fuzzy measure and integrator design in recursive aggregation:

*Step 1:* According to the information evaluation table, we utilize the genetic algorithm to gain $\lambda$ and $g_\lambda(\{x_j\})$.

*Step 2:* Evaluation function:

$$Eval(V) = \min\{\frac{1}{m}\sum_{i=1}^m D^2(\tilde{F}_i(\tilde{\alpha}_{i1}, \tilde{\alpha}_{i2}, \cdots, \tilde{\alpha}_{ik}), \tilde{E}_i)\}$$

s.t

$$
\begin{cases}
\mu_{\tilde{F}_i} = 1 - \prod_{j=1}^k (1 - \mu_{\tilde{\alpha}_{ij}})^{g_\lambda(A_j) - g_\lambda(A_{j-1})}, \\
v_{\tilde{F}_i} = \prod_{j=1}^k v_{\tilde{\alpha}_{ij}}^{g_\lambda(A_j) - g_\lambda(A_{j-1})}, \\
\pi_{\tilde{F}_i} = 1 - \mu_{\tilde{F}_i} - v_{\tilde{F}_i}, \\
1 + \lambda = \prod_{j=1}^k (1 + \lambda g_\lambda^{(k)}(x_j)), \\
g_\lambda^{(k)}(A_j) = \frac{1}{\lambda}(\prod_{x \in A_j}(1 + \lambda g_\lambda^{(k)}(\{x\})) - 1).
\end{cases}
$$

*Step 3:* Employ $g_\lambda^{(k)}(A_j) = \frac{1}{\lambda}(\prod_{x \in A_j}(1 + \lambda g_\lambda^{(k)}(\{x\})) - 1)$ to derive $g_\lambda^{(k)}(A_j)$.

*Step 4:* Utilize RA for IFNs based on a NAM with $\sigma - \lambda$ rules proposed in this work to obtain $g_\lambda^{(k+1)}(A_j)$ with the condition of increasing the attribute index $x_{k+1}$.

*Step 5:* When a new object $i$ gives values to the $k + 1$ attributes, we can utilize the presented OWA operator to obtain $\tilde{F}_{k(i)}$. Then, utilize RF for IFNs based on a NAM with

$\sigma - \lambda$ rules to directly obtain aggregation results $\tilde{F}_{k+1(i)}$ of $k + 1$ attributes from aggregation results $\tilde{F}_{k(i)}$ of $k$ attributes.

*Step 6:* Similarly, when adding, in turn, the attribute index to evaluation, we can always utilize the old aggregation results $\tilde{F}_{n-1(i)}$ of $n - 1$ attributes to directly derive the final aggregation results $\tilde{F}_{n(i)}$ of $n$ attributes.

**Example 1.** *Online shopping is prevalent in e-commercial area. Thus, making a relatively accurate and reasonable evaluation for online shopping is very useful. An online shop's management randomly chooses five customers for a satisfaction evaluation of online shop. The evaluation value is denoted as $\tilde{a}_{ij}$, and $\tilde{a}_{ij}$ is an IFN. The attributes are $x_1$: logistics, $x_2$: service attitude, $x_3$: price, $x_4$: product quality in satisfaction evaluation, and the results are shown in Table 2. When customers further consider "after-sale service" or "payment security", the online shop management want to obtain a new evaluation result.*

*Step 1:* Collecting the evaluation values of the customers for a good in Table 2. The IFNs in Table 2 shows that the evaluation values of the customers for the attributes of the good. That is to say, the satisfaction evaluation and dissatisfaction evaluation of the customer for the attribute of a good.

*Step 2:* According to Table 3, we utilize the genetic algorithm to gain $\lambda$ and $g_\lambda^{(4)}(\{x_j\})(j = 1, 2, 3, 4)$. The genetic algorithm is shown in Algorithm 1.

---

**Algorithm 1:** The genetic algorithm to gain $\lambda$ and $g_\lambda^{(4)}(\{x_j\})(j = 1, 2, 3, 4)$.

*Step 1:* Build an evaluation function file in Matlab software, save it as fit.m file and store it in the appropriate directory.

*Step 2:* Genetic algorithm toolbox is executed in the command window of Matlab R2014a to enter the GUI interface of the genetic algorithm and set relevant parameters in the corresponding column. Fitness function is @fit, variables numbers are five. The parameters of the genetic algorithm used in this paper are shown in Table 4. Stopping Generation is 149, and we use the default values for others. The GUI running interface of the genetic algorithm toolbox is shown in Figure 1.

*Step 3:* After setting the parameters, click "Start" in the GUI running interface of the genetic algorithm toolbox. $\lambda$ and $g_\lambda^{(4)}(\{x_j\})(j = 1, 2, 3, 4)$ are shown in Table 5, and the operation result diagram is shown in Figure 2.

---

*Step 3:* Utilizing $g_\lambda^{(4)}(A_j) = \frac{1}{\lambda}(\prod_{x \in A_j}(1 + \lambda g_\lambda^{(4)}(\{x\})) - 1)$ to derive $g_\lambda^{(4)}(A_j)$.

$$g_\lambda^{(4)}(A_1) = g_\lambda^{(4)}(\{x_1\}) = 0.3740; \quad g_\lambda^{(4)}(A_2) = \frac{1}{\lambda}[(1 + \lambda g_\lambda^{(4)}(\{x_1\}))(1 + \lambda g_\lambda^{(4)}(\{x_2\})) - 1] = 0.6335;$$
$$g_\lambda^{(4)}(A_3) = \frac{1}{\lambda}[(1 + \lambda g_\lambda^{(4)}(\{x_1\}))(1 + \lambda g_\lambda^{(4)}(\{x_2\}))(1 + \lambda g_\lambda^{(4)}(\{x_3\})) - 1] = 0.7477.$$

*Step 4:* Utilizing LRF for IFNs based on a NAM with $\sigma - \lambda$ rules proposed in this paper to obtain $g_\lambda^{(5)}(A_j)$ with the attribute index "after-sale service" increasing, we have

$$\alpha = orness(g_\lambda^{(4)}) = \frac{1}{3}\sum_{j=1}^{3} g_\lambda^{(4)}(A_j) = 0.5850; \quad P_L^{(5)} = \frac{(n-1)\alpha}{(n-2)\alpha+1} = \frac{4\alpha}{3\alpha+1} = 0.8494;$$

$$g_\lambda^{(5)}(A_1) = P_L^{(5)} \cdot g_\lambda^{(4)}(A_1) = 0.3177; \quad g_\lambda^{(5)}(A_2) = P_L^{(5)} \cdot g_\lambda^{(4)}(A_2) = 0.5381;$$
$$g_\lambda^{(5)}(A_3) = P_L^{(5)} \cdot g_\lambda^{(4)}(A_3) = 0.6351; \quad g_\lambda^{(5)}(A_4) = P_L^{(5)} = 0.8494.$$

*Step 5:* When a new customer (6) gives values to logistics, service attitude, price, product quality, as listed in Table 2. We can utilize the OWA operator proposed in this paper to obtain $\tilde{F}_{4(6)}$. Then, utilize LRF to directly derive aggregation results $\tilde{F}_{5(6)}$ of 5 attributes from aggregation results $\tilde{F}_{4(6)}$ of 4 attributes.

$$\tilde{F}_{4(6)} = (g_\lambda^{(4)}(A_1)) \otimes (0.90, 0.60) \oplus (g_\lambda^{(4)}(A_2) - g_\lambda^{(4)}(A_1)) \otimes (0.60, 0.20) \oplus (g_\lambda^{(4)}(A_3) - g_\lambda^{(4)}(A_2)) \otimes (0.30, 0.60) \oplus (1 - g_\lambda^{(4)}(A_3)) \otimes (0.20, 0.70) = (0.6975, 0),$$

$$\tilde{F}_{5(6)} = P_L^{(5)} \otimes \tilde{F}_{4(6)} \oplus (1 - g_\lambda^{(5)}(A_4)) \otimes (0.40, 0.50) = (0.6646, 0).$$

**Step 6:** When a new customer (7) gives values to the attributes, as listed in Table 2,

$$\tilde{F}_{6(7)} = P_L^{(6)} \otimes \tilde{F}_{5(7)} \oplus (1 - g_\lambda^{(6)}(A_5)) \otimes (0.50, 0.45) = (0.6476, 0.2989).$$

**Table 2.** The satisfaction evaluation.

| Customer | Logistics | Service Attitude | Price | Product Quality | After-Sale Service |
|---|---|---|---|---|---|
| 1 | $(0.00, 0.90)$ | $(0.10, 0.80)$ | $(0.30, 0.50)$ | $(1.00, 0.00)$ | |
| 2 | $(0.30, 0.60)$ | $(0.80, 0.15)$ | $(0.50, 0.30)$ | $(0.60, 0.30)$ | |
| 3 | $(0.70, 0.20)$ | $(0.90, 0.05)$ | $(1.00, 0.00)$ | $(0.70, 0.2)$ | |
| 4 | $(0.10, 0.85)$ | $(0.00, 0.90)$ | $(0.40, 0.50)$ | $(0.30, 0.50)$ | |
| 5 | $(0.50, 0.35)$ | $(0.40, 0.40)$ | $(0.50, 0.50)$ | $(0.60, 0.20)$ | |
| 6 | $(0.30, 0.60)$ | $(0.90, 0.00)$ | $(0.60, 0.20)$ | $(0.20, 0.70)$ | $(0.40, 0.50)$ |
| 7 | $(0.30, 0.70)$ | $(0.90, 0.10)$ | $(0.60, 0.20)$ | $(0.20, 0.70)$ | $(0.40, 0.60)$ |

| Customer | Payment Security | Evaluation |
|---|---|---|
| 1 | | $(0.20, 0.65)$ |
| 2 | | $(0.50, 0.45)$ |
| 3 | | $(0.80, 0.10)$ |
| 4 | | $(0.10, 0.85)$ |
| 5 | | $(0.40, 0.50)$ |
| 6 | | ? |
| 7 | $(0.50, 0.45)$ | ? |

**Table 3.** The triangular fuzzy number changed into the interval number.

| Customer | Logistics | Service Attitude | Price | Product Quality | Evaluation |
|---|---|---|---|---|---|
| 1 | $(0.00, 0.90)$ | $(0.10, 0.80)$ | $(0.30, 0.50)$ | $(1.00, 0.00)$ | $(0.20, 0.65)$ |
| 2 | $(0.30, 0.60)$ | $(0.80, 0.15)$ | $(0.50, 0.30)$ | $(0.60, 0.30)$ | $(0.50, 0.45)$ |
| 3 | $(0.70, 0.20)$ | $(0.90, 0.05)$ | $(1.00, 0.00)$ | $(0.70, 0.20)$ | $(0.80, 0.10)$ |
| 4 | $(0.10, 0.85)$ | $(0.00, 0.90)$ | $(0.40, 0.50)$ | $(0.30, 0.50)$ | $(0.10, 0.85)$ |
| 5 | $(0.50, 0.35)$ | $(0.40, 0.40)$ | $(0.50, 0.50)$ | $(0.60, 0.20)$ | $(0.40, 0.50)$ |

**Table 4.** The parameters of the genetic algorithm.

| Parameter | Population Size | Initial Rang | Elite Count | Crossover Probability |
|---|---|---|---|---|
| | 80 | $[0; 1]$ | 2 | 0.75 |

**Table 5.** Results of Step 2.

| Point Set | $\lambda$ | $g_\lambda$ | Error | Hereditary Algebra |
|---|---|---|---|---|
| $\{x_1\}$ | 0.0160 | 0.3740 | 0.0040 | 149 |
| $\{x_2\}$ | | 0.2580 | | |
| $\{x_3\}$ | | 0.1130 | | |
| $\{x_4\}$ | | 0.0020 | | |

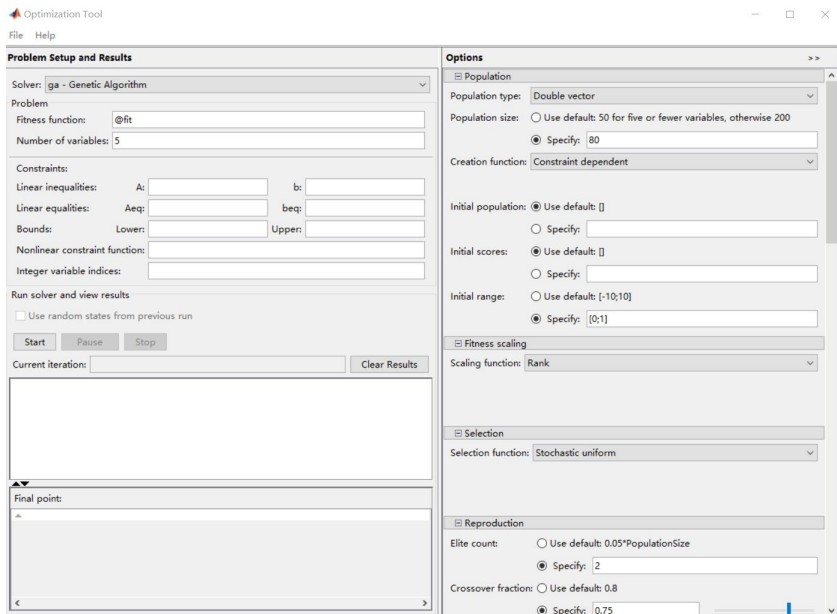

**Figure 1.** The GUI running interface of the genetic algorithm toolbox.

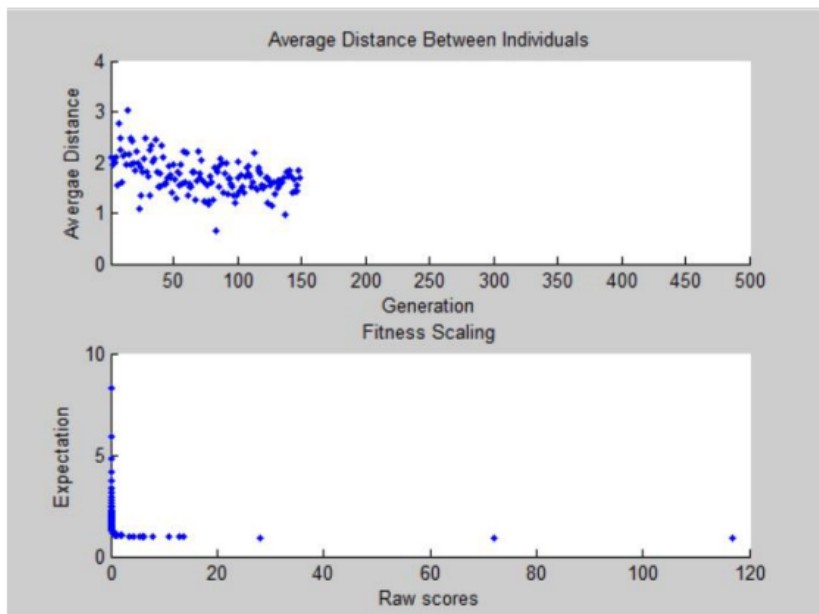

**Figure 2.** The operation result diagram of Step 2.

## 6. Conclusions

In this paper, we propose an OWA operator for IFN based on a NAM with $\sigma - \lambda$ rules and derive the Recursive Forms of OWA operators for IFN according to NAM with $\sigma - \lambda$ rules while keeping the orness grade unchanged. Furthermore, the procedure for integrator design in recursive aggregation is designed and the process of calculation is demonstrated by an example.

**Author Contributions:** Conceptualization, Z.G.; investigation, Y.S. All authors have read and agreed to the published version of the manuscript.

**Funding:** This research was funded by the National Natural Science Foundation of China (Grant No. 12061067).

**Data Availability Statement:** Not applicable.

**Acknowledgments:** The authors would like to thank the referees for providing very helpful comments and suggestions.

**Conflicts of Interest:** The authors declare no conflict of interest.

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
