# Peer review of "Recursive Aggregation and Its Fusion Process for Intuitionistic Fuzzy Numbers Based on Non-Additive Measure"

_axioms, doi:10.3390/axioms11060257_

Round 1

Reviewer 1 Report

The authors' article is devoted to the development and refinement of Yager's theory regarding the improvement of the recursive algorithm for calculating ordered weighted averaging (OWA). The authors take Yager's idea as a basis and use recursive forms of OWA operators, but choose intuitionistic fuzzy numbers (IFN) as a basis. In the article, the authors introduce a number of definitions, formulate statements and prove them. Then, an algorithm is developed for applying ntuitionistic fuzzy numbers within the framework of the OWA calculation method. In the last section of the work, an example of the application of the author's methodology with sequential steps (calculations) is given. The article contains all the necessary links to the works of scientists specializing in this field of knowledge.
Remark 1. The algorithm of the authors is quite complicated for perception, therefore, in my opinion, it is necessary to carry out a number of intermediate calculations in more detail. This will increase the understanding of the material for the readers of the magazine.
Remark 2. There are a number of misprints in the work. They need to be corrected. In Definition 3.1 and Theorem 3.1 check the use of indexes i=1..n (Ai and Ai-1) .
Remark 3. In the example given by the authors, it is not clear why the input variables are defined in this way from the very beginning. Where did the ranges come from? How are these variables consistent with the economic meaning of the example under consideration?

Author Response

refer to attached file Revision Note of axioms-1692569 Revised.

Reviewer 2 Report

A study on the recursive aggregation of OWA operators for intuitionistic fuzzy numbers is presented.

The introduction is very poor. The authors must discuss in greater detail the state of the art on the ticursive forms of OWA operators, the limits of current studies and the motivations and objectives of their research.

Examples of OWA operator for IFNs and recursive aggregation for OWA operators and need to be added in sections 3 and 4.

The procedure for the calculation of fuzzy measure and integrator design in recursive aggregation should be discussed in more detail and all the indices in the formulas of the various steps must be well specified.

In addition, the parameters of the genetic algorithm used to gain λ and gλ (composition of genes and chromosomes, fitness function, type of genetic operators used for selection, crossover and mutation, etc.) must be detailed.

English quality is  poor. Please proofread.

Author Response

(The authors gave the same response as above.)

Reviewer 3 Report

This paper proposes a recursive function in order to aggregate OWA operators for IFN. The authors claim to keep the orness of the aggregator and they provide an algorithmic solution to obtain their results. The idea of the paper is fair and the proposal is original. However, the manuscript lacks more formal definitions expected in this kind of work. Please refer to the following comments:

  1. Def. 2.1: the measure function (\mu) is not clearly defined, as it does not have the domain and codomain sets. This is recommended for the definitions of all functions throughout the paper.
  2. Def. 2.4: score functions are defined twice. It seems H(\alpha) should be an accuracy function.
  3. Def. 2.5: The operation law is not clearly defined. The $\otimes$ and $\oplus$ operators were not defined accordingly, as seen in ref. [11].

The paper also contains an extensive list of English grammar errors and typos. To name a few, note that acronyms with vowel sounds should have "an" as the indefinite article; the verb to "be" is used frequently but in some cases, it should be "is" / "are"; it appears that the word "Reft" is used to refer to "Right".

A major review is needed in Section 5, since it is a bit too informal and hard to read. For the readers, probably it would be interesting to include more details about the procedure presented. The example should also be better explained and the algorithm should be formally provided. 

Finally, the references seems to be a bit outdated.  It is not clear in the manuscript if the authors verified the recent works on orness, and the recent trends in aggregation/fusion functions. This discussion  should be addressed.

Author Response

We have made a thorough revision for our paper according to the comments of  yours. Main changes made to the revised manuscript and reply to you and a summary of the revisions are shown in the attached file: Revision Note of axioms-1692569 Revised 2nd round, and in highlight of axioms-1692569 Revised 2nd round.

Round 2

Reviewer 2 Report

The authors carefully revised their paper taking into account all my suggestions. I consider this paper publishable in the present form.

Author Response

Thank you very much for your review opinion "The authors carefully revised their paper taking into account all my suggestions. I consider this paper publishable in the present form."

Reviewer 3 Report

The authors performed a good job in order to improve the manuscript that should be accepted in the current form.